# Kernel functions based on triplet comparisons

**Matthäus Kleindessner**[*]
Department of Computer Science
Rutgers University
Piscataway, NJ 08854
mk1572@cs.rutgers.edu

**Ulrike von Luxburg**
Department of Computer Science
University of Tübingen
Max Planck Institute for Intelligent Systems, Tübingen
luxburg@informatik.uni-tuebingen.de

## Abstract

Given only information in the form of similarity triplets "Object A is more similar to object B than to object C" about a data set, we propose two ways of defining a kernel function on the data set. While previous approaches construct a low-dimensional Euclidean embedding of the data set that reflects the given similarity triplets, we aim at defining kernel functions that correspond to high-dimensional embeddings. These kernel functions can subsequently be used to apply any kernel method to the data set.

## 1   Introduction

Assessing similarity between objects is an inherent part of many machine learning problems, be it in an unsupervised task like clustering, in which similar objects should be grouped together, or in classification, where many algorithms are based on the assumption that similar inputs should produce similar outputs. In a typical machine learning setting one assumes to be given a data set $\mathcal{D}$ of objects together with a dissimilarity function $d$ (or, equivalently, a similarity function $s$) quantifying how "close" objects are to each other. In recent years, however, a new branch of the machine learning literature has emerged that relaxes this scenario (see the next paragraph and Section 3 for references). Instead of being able to evaluate $d$ itself, we only get to see a collection of similarity triplets of the form "Object $A$ is more similar to object $B$ than to object $C$", which claims that $d(A, B) < d(A, C)$. The main motivation for this relaxation comes from human-based computation: It is widely accepted that humans are better and more reliable at providing similarity triplets, which means assessing similarity on a relative scale, than at providing similarity estimates on an absolute scale ("The similarity between objects $A$ and $B$ is 0.8"). This can be seen as a special case of the general observation that humans are better at comparing two stimuli than at identifying a single one (Stewart et al., 2005). For this reason, whenever one is lacking a meaningful dissimilarity function that can be evaluated automatically and has to incorporate human expertise into the machine learning process, collecting similarity triplets (e.g., via crowdsourcing) may be an appropriate means.

Given a data set $\mathcal{D}$ and similarity triplets for its objects, it is not immediately clear how to solve machine learning problems on $\mathcal{D}$. A general approach is to construct an ordinal embedding of $\mathcal{D}$, that is to map objects to a Euclidean space of a small dimension such that the given triplets are preserved as well as possible (Agarwal et al., 2007; Tamuz et al., 2011; van der Maaten and Weinberger, 2012; Terada and von Luxburg, 2014; Amid and Ukkonen, 2015; Heim et al., 2015; Amid et al., 2016; Jain et al., 2016). Once such an ordinal embedding has been constructed, one can solve a problem on $\mathcal{D}$ by solving it on the embedding. Only recently, algorithms have been proposed for solving various specific problems directly without constructing an ordinal embedding as an intermediate step (Heikinheimo and Ukkonen, 2013; Kleindessner and von Luxburg, 2017). With this paper we provide another generic means for solving machine learning problems based on similarity triplets that is different from

---

[*]Work done while being a PhD student at the University of Tübingen.

the ordinal embedding approach. We define two data-dependent kernel functions on $\mathcal{D}$, corresponding to high-dimensional embeddings of $\mathcal{D}$, that can subsequently be used by any kernel method. Our proposed kernel functions measure similarity between two objects in $\mathcal{D}$ by comparing to which extent the two objects give rise to resembling similarity triplets. The intuition is that this quantifies the relative difference in the locations of the two objects in $\mathcal{D}$. Experiments on both artificial and real data show that this is indeed the case and that the similarity scores defined by our kernel functions are meaningful. Our approach is appealingly simple, and other than ordinal embedding algorithms our kernel functions are deterministic and parameter-free. We observe them to run significantly faster than well-known embedding algorithms and to be ideally suited for a landmark design.

**Setup**    Let $\mathcal{X}$ be an arbitrary set and $d : \mathcal{X} \times \mathcal{X} \to \mathbb{R}_0^+$ be a symmetric dissimilarity function on $\mathcal{X}$: a higher value of $d$ means that two elements of $\mathcal{X}$ are more dissimilar to each other. The terms dissimilarity and distance are used synonymously. To simplify presentation, we assume that for all triples of distinct objects $A, B, C \in \mathcal{X}$ either $d(A, B) < d(A, C)$ or $d(A, B) > d(A, C)$ is true. Note that we do not require $d$ to be a metric. We formally define a similarity triplet as binary answer to a dissimilarity comparison

$$d(A, B) \overset{?}{<} d(A, C). \tag{1}$$

We refer to $A$ as the anchor object. A similarity triplet can be incorrect, meaning that it claims a positive answer to the comparison (1) although in fact the negative answer is true. In the following, we deal with a finite data set $\mathcal{D} = \{x_1, \ldots, x_n\} \subseteq \mathcal{X}$ and collections of similarity triplets that are encoded as follows: an ordered triple of distinct objects $(x_i, x_j, x_k)$ means $d(x_i, x_j) < d(x_i, x_k)$. A collection of similarity triplets is the only information that we are given about $\mathcal{D}$. Note that such a collection does not necessarily provide an answer to every possible dissimilarity comparison (1).

## 2    Our kernel functions

Assume we are given a collection $\mathcal{S}$ of similarity triplets for the objects of $\mathcal{D}$. Similarity triplets in $\mathcal{S}$ can be incorrect, but for the moment assume that contradicting triples $(x_i, x_j, x_k)$ and $(x_i, x_k, x_j)$ cannot be present in $\mathcal{S}$ at the same time. We will discuss how to deal with the general case below.

**Kernel function $k_1$**    Our first kernel function is based on the following idea: We fix two objects $x_a$ and $x_b$. In order to compute a similarity score between $x_a$ and $x_b$ we would like to rank all objects in $\mathcal{D}$ with respect to their distance from $x_a$ and also rank them with respect to their distance from $x_b$, and take a similarity score between these two rankings as similarity score between $x_a$ and $x_b$. One possibility to measure similarity between rankings is given by the famous Kendall tau correlation coefficient (Kendall, 1938), which is also known as Kendall's $\boldsymbol{\tau}$: for two rankings of $n$ items, Kendall's $\boldsymbol{\tau}$ between the two rankings is the fraction of concordant pairs of items minus the fraction of discordant pairs of items. Here, a pair of two items $i_1$ and $i_2$ is concordant if $i_1 \prec i_2$ or $i_1 \succ i_2$ according to both rankings, and discordant if it satisfies $i_1 \prec i_2$ according to one and $i_1 \succ i_2$ according to the other ranking. Formally, a ranking is represented by a permutation $\sigma : \{1, \ldots, n\} \to \{1, \ldots, n\}$ such that $\sigma(i) \neq \sigma(j)$, $i \neq j$, and $\sigma(i) = m$ means that item $i$ is ranked at the $m$-th position. Given two rankings $\sigma_1$ and $\sigma_2$, the number of concordant pairs equals

$$f_c(\sigma_1, \sigma_2) = \sum_{i<j} [\mathbb{1}\{\sigma_1(i) < \sigma_1(j)\}\mathbb{1}\{\sigma_2(i) < \sigma_2(j)\} + \mathbb{1}\{\sigma_1(i) > \sigma_1(j)\}\mathbb{1}\{\sigma_2(i) > \sigma_2(j)\}],$$

the number of discordant pairs equals

$$f_d(\sigma_1, \sigma_2) = \sum_{i<j} [\mathbb{1}\{\sigma_1(i) < \sigma_1(j)\}\mathbb{1}\{\sigma_2(i) > \sigma_2(j)\} + \mathbb{1}\{\sigma_1(i) > \sigma_1(j)\}\mathbb{1}\{\sigma_2(i) < \sigma_2(j)\}],$$

and Kendall's $\boldsymbol{\tau}$ between $\sigma_1$ and $\sigma_2$ is given by $\boldsymbol{\tau}(\sigma_1, \sigma_2) = [f_c(\sigma_1, \sigma_2) - f_d(\sigma_1, \sigma_2)] / \binom{n}{2}$.

By measuring similarity between the two rankings of objects (one with respect to their distance from $x_a$ and one with respect to their distance from $x_b$) with Kendall's $\boldsymbol{\tau}$ we would compute a similarity score between $x_a$ and $x_b$. This idea is illustrated with an example in Figure 1 (left). It has been established recently that Kendall's $\boldsymbol{\tau}$ is actually a kernel function on the set of total rankings (Jiao and Vert, 2015). Hence, by measuring similarity on $\mathcal{D}$ in the described way we would even end up with a

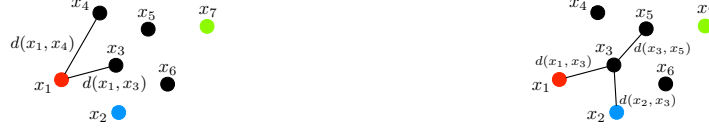

Figure 1: Illustrations of the ideas behind $k_1$ (left) and $k_2$ (right). **For $k_1$:** In order to compute a similarity score between $x_1$ (in red) and $x_2$ (in blue) we would like to rank all objects with respect to their distance from $x_1$ and also with respect to their distance from $x_2$ and compute Kendall's $\tau$ between the two rankings. In this example, the objects would rank as $x_1 \prec x_3 \prec x_2 \prec x_4 \prec x_5 \prec x_6 \prec x_7$ and $x_2 \prec x_3 \prec x_6 \prec x_1 \prec x_5 \prec x_4 \prec x_7$, respectively. Kendall's $\tau$ between these two rankings is $1/3$, and this would be the similarity score between $x_1$ and $x_2$. For comparison, the score between $x_1$ and $x_7$ (in green) would be $-5/7$, and between $x_2$ and $x_7$ it would be $-3/7$. **For $k_2$:** In order to compute a similarity score between $x_1$ and $x_2$ we would like to check for every pair of objects $(x_i, x_j)$ whether the distance comparisons $d(x_i, x_1) \overset{?}{<} d(x_i, x_j)$ and $d(x_i, x_2) \overset{?}{<} d(x_i, x_j)$ yield the same result or not. Here, we have 32 pairs for which they yield the same result and 17 pairs for which they do not. We would assign $7^{-2} \cdot (32 - 17) = 15/49$ as similarity score between $x_1$ and $x_2$. The score between $x_1$ and $x_7$ would be $3/49$, and between $x_2$ and $x_7$ it would be $1/49$.

kernel function on $\mathcal{D}$ since the following holds: for any mapping $h : \mathcal{D} \to \mathcal{Z}$ and kernel function $k : \mathcal{Z} \times \mathcal{Z} \to \mathbb{R}$, $k \circ (h, h) : \mathcal{D} \times \mathcal{D} \to \mathbb{R}$ is a kernel function.

In our situation, the problem is that in most cases $\mathcal{S}$ will contain only a small fraction of all possible similarity triplets and also that some of the triplets in $\mathcal{S}$ might be incorrect, so that there is no way of ranking all objects with respect to their distance from any fixed object based on the similarity triplets in $\mathcal{S}$. To adapt the procedure, we consider a feature map that corresponds to the kernel function just described. By a feature map corresponding to a kernel function $k : \mathcal{D} \times \mathcal{D} \to \mathbb{R}$ we mean a mapping $\Phi : \mathcal{D} \to \mathbb{R}^m$ for some $m \in \mathbb{N}$ such that $k(x_i, x_j) = \langle \Phi(x_i), \Phi(x_j) \rangle = \Phi(x_i)^T \cdot \Phi(x_j)$. It is easy to see from the above formulas (also compare with Jiao and Vert, 2015) that a feature map corresponding to the described kernel function is given by $\Phi_{k_\tau} : \mathcal{D} \to \mathbb{R}^{\binom{n}{2}}$ with

$$\Phi_{k_\tau}(x_a) = \frac{1}{\sqrt{\binom{n}{2}}} \cdot \left( \mathbb{1}\{d(x_a, x_i) < d(x_a, x_j)\} - \mathbb{1}\{d(x_a, x_i) > d(x_a, x_j)\} \right)_{1 \leq i < j \leq n}.$$

In our situation, where we are only given $\mathcal{S}$ and cannot evaluate $\Phi_{k_\tau}$ in most cases, we have to replace $\Phi_{k_\tau}$ by an approximation: up to a normalizing factor, we replace an entry in $\Phi_{k_\tau}(x_a)$ by zero if we cannot evaluate it based on the triplets in $\mathcal{S}$. More precisely, we consider the feature map $\Phi_{k_1} : \mathcal{D} \to \mathbb{R}^{\binom{n}{2}}$ given by $\Phi_{k_1}(x_a) = ([\Phi_{k_1}(x_a)]_{i,j})_{1 \leq i < j \leq n}$ with

$$[\Phi_{k_1}(x_a)]_{i,j} = \frac{1}{\sqrt{|\{(x_i, x_j, x_k) \in \mathcal{S} : x_i = x_a\}|}} \cdot \left( \mathbb{1}\{(x_a, x_i, x_j) \in \mathcal{S}\} - \mathbb{1}\{(x_a, x_j, x_i) \in \mathcal{S}\} \right) \tag{2}$$

and define our first proposed kernel function $k_1 : \mathcal{D} \times \mathcal{D} \to \mathbb{R}$ by

$$k_1(x_i, x_j) = \Phi_{k_1}(x_i)^T \cdot \Phi_{k_1}(x_j). \tag{3}$$

Note that the scaling factor in the definition of $\Phi_{k_1}$, ensuring that the feature embedding lies on the unit sphere, is crucial whenever the number of similarity triplets in which an object appears as anchor object is not approximately constant over the different objects. For ease of exposition we have assumed that every object in $\mathcal{D}$ appears at least once as an anchor object in a similarity triplet in $\mathcal{S}$. In the unlikely case that $x_a$ does not appear at least once as an anchor object, meaning that we do not have any information for ranking the objects in $\mathcal{D}$ with respect to their distance from $x_a$ at all, we simply set $\Phi_{k_1}(x_a)$ to zero (which is consistent with (2) under the convention "0/0=0").

**Kernel function $k_2$** Our second kernel function is based on a similar idea. Now we do not consider $x_a$ and $x_b$ as anchor objects when measuring their similarity, but compare whether they rank similarly with respect to their distances from the various other objects. Concretely, we would like to count the number of pairs of objects $(x_i, x_j)$ for which the comparisons

$$d(x_i, x_a) \overset{?}{<} d(x_i, x_j) \quad \text{and} \quad d(x_i, x_b) \overset{?}{<} d(x_i, x_j) \tag{4}$$

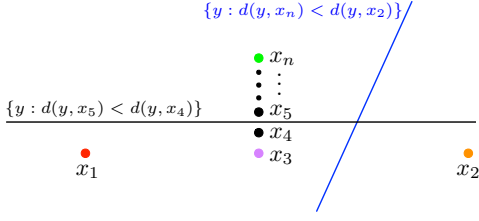

Figure 2: $k_1$ measures similarity between two objects by counting in how many of the halfspaces that are obtained from distance comparisons the two objects reside at the same time. The outcome does not only depend on the distance between the two objects, but also on their location within the data set: although $x_1$ and $x_2$ are located far apart, $k_1$ considers them to be very similar. See the running text for details.

yield the same result and subtract the number of pairs for which these comparisons yield different results. See the right-hand side of Figure 1 for an illustration of this idea. Adapted to our situation of being only given $\mathcal{S}$ it corresponds to considering the feature map $\Phi_{k_2} : \mathcal{D} \to \mathbb{R}^{n^2}$ given by

$$\Phi_{k_2}(x_a) = \frac{1}{\sqrt{|\{(x_i, x_j, x_k) \in \mathcal{S} : x_j = x_a \vee x_k = x_a\}|}} \cdot$$

$$\left( \mathbb{1}\{(x_i, x_a, x_j) \in \mathcal{S}\} - \mathbb{1}\{(x_i, x_j, x_a) \in \mathcal{S}\} \right)_{1 \le i,j \le n}$$

and defining our second proposed kernel function $k_2 : \mathcal{D} \times \mathcal{D} \to \mathbb{R}$ by

$$k_2(x_i, x_j) = \Phi_{k_2}(x_i)^T \cdot \Phi_{k_2}(x_j). \tag{5}$$

Again, the scaling factor in the definition of $\Phi_{k_2}$ is crucial whenever there are objects appearing in more similarity triplets than others and we apply the convention "0/0=0".

**Contradicting similarity triplets** If $\mathcal{S}$ contains contradicting triples $(x_i, x_j, x_k)$ and $(x_i, x_k, x_j)$ and there might be triples being present repeatedly, one can alter the definition of $\Phi_{k_1}$ or $\Phi_{k_2}$ as follows: if $\#\{(x_a, x_i, x_j) \in \mathcal{S}\}$ denotes the number of how often the triple $(x_a, x_i, x_j)$ appears in $\mathcal{S}$, set $\Phi_{k_1}(x_a) = \widetilde{\Phi}_{k_1}(x_a) / \|\widetilde{\Phi}_{k_1}(x_a)\|$ where $\widetilde{\Phi}_{k_1}(x_a)$ equals

$$\left( \frac{\#\{(x_a, x_i, x_j) \in \mathcal{S}\} - \#\{(x_a, x_j, x_i) \in \mathcal{S}\}}{\#\{(x_a, x_i, x_j) \in \mathcal{S}\} + \#\{(x_a, x_j, x_i) \in \mathcal{S}\}} \right)_{1 \le i < j \le n}.$$

The definition of $\Phi_{k_2}$ can be revised in an analogous way. In doing so, we incorporate a simple estimate of the likelihood of a triple being correct.

## 2.1 Reducing diagonal dominance

If the number $|\mathcal{S}|$ of given similarity triplets is small, our kernel functions suffer from a problem that is shared by many other kernel functions defined on complex data: $\Phi_{k_1}$ and $\Phi_{k_2}$ map the objects in $\mathcal{D}$ to sparse vectors, that is almost all of their entries are zero. As a consequence, two different feature vectors $\Phi_{k_i}(x_a)$ and $\Phi_{k_i}(x_b)$ appear to be almost orthogonal and the similarity score $k_i(x_a, x_b)$ is much smaller than the self-similarity scores $k_i(x_a, x_a)$ or $k_i(x_b, x_b)$. This phenomenon, usually referred to as diagonal dominance of the kernel function, has been observed to pose difficulties for the kernel methods using the kernel function, and several ways have been proposed for dealing with it (Schölkopf et al., 2002; Greene and Cunningham, 2006). In all our experiments we deal with diagonal dominance in the following simple way: Let $k$ denote a kernel function and $K$ the kernel matrix on $\mathcal{D}$, that is $K = (k(x_i, x_j))_{i,j=1}^n$, which would be the input to a kernel method. Then we replace $K$ by $K - \lambda_{\min} I$ where $I \in \mathbb{R}^{n \times n}$ denotes the identity matrix and $\lambda_{\min}$ is the smallest eigenvalue of $K$.

## 2.2 Geometric intuition

Intuitively, our kernel functions measure similarity between $x_a$ and $x_b$ by quantifying to which extent $x_a$ and $x_b$ can be expected to be located in the same region of $\mathcal{D}$: Think of $\mathcal{D}$ as a subset of $\mathbb{R}^m$ and $d$ being the Euclidean metric. A similarity triplet $d(x_a, x_i) < d(x_a, x_j)$ then tells us that $x_a$ resides in the halfspace defined by the hyperplane that is perpendicular to the line segment connecting $x_i$ and $x_j$ and goes through the segment's midpoint. If there is also a similarity triplet $d(x_b, x_i) < d(x_b, x_j)$, $x_a$ and $x_b$ thus are located in the same halfspace (assuming the correctness of the similarity triplets) and this is reflected by a higher value of $k_1(x_a, x_b)$. Similarly, a similarity triplet $d(x_i, x_a) < d(x_i, x_j)$

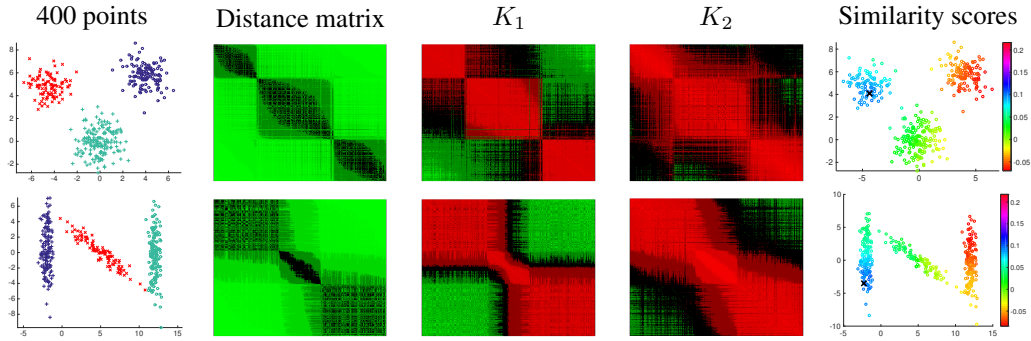

Figure 3: Kernel matrices for two data sets, each consisting of 400 points, based on 10% of all similarity triplets. 1st plot of a row: Data points. 2nd plot: Distance matrix. 3rd / 4th plot: Kernel matrix for $k_1$ / $k_2$. 6th plot: Similarity scores between a fixed point and the other points (for $k_1$).

tells us that $x_a$ is located in a ball with radius $d(x_i, x_j)$ centered at $x_i$, and the value of $k_2(x_a, x_b)$ is higher if there is a similarity triplet $d(x_i, x_b) < d(x_i, x_j)$ telling us that $x_b$ is located in this ball too and it is smaller if there is a triplet $d(x_i, x_j) < d(x_i, x_b)$ telling us that $x_b$ is not located in this ball.

Note that the similarity scores between $x_a$ and $x_b$ defined by $k_1$ or $k_2$ do not only depend on $d(x_a, x_b)$, but rather on the locations of $x_a$ and $x_b$ within $\mathcal{D}$ and on how the points in $\mathcal{D}$ are spread in the space since this affects how the various hyperplanes or balls are related to each other. Consider the example illustrated in Figure 2: Let $d(x_3, x_n) = 1$ implying that $d(x_i, x_{i+1}) = \Theta(1/n)$, $3 \leq i < n$, and $d(x_1, x_2) > d(x_2, x_n) > d(x_1, x_n) > d(x_2, x_3) > d(x_1, x_3) > 1$ be arbitrarily large. Although $x_1$ and $x_2$ are located at the maximum distance to each other, they satisfy $d(x_1, x_i) < d(x_1, x_j)$ and $d(x_2, x_i) < d(x_2, x_j)$ for all $3 \leq i < j \leq n$, and hence both $x_1$ and $x_2$ are jointly located in all the halfspaces obtained from these distance comparisons. We end up with $k_1(x_1, x_2) \to 1$, $n \to \infty$, assuming $k_1$ is computed based on all possible similarity triplets, all of which are correct. The distance between $x_3$ and $x_n$ is much smaller, but there are many points in between them and the hyperplanes obtained from the distance comparisons with these points separate $x_3$ and $x_n$. We end up with $k_1(x_3, x_n) \to -1$, $n \to \infty$. Depending on the task at hand, this may be desirable or not.

Let us examine the meaningfulness of our kernel functions by calculating them on five visualizable data sets. Each of the first four data sets consists of 400 points in $\mathbb{R}^2$ and $d$ equals the Euclidean metric. The fifth data set consists of 400 vertices of an undirected graph from a stochastic block model and $d$ equals the shortest path distance. We computed $k_1$ and $k_2$ based on 10% of all possible similarity triplets (chosen uniformly at random from all triplets). The results for the first two data sets are shown in Figure 3. The results for the remaining data sets are shown in Figure 6 in Section A.1 in the supplementary material. The first plot of a row shows the data set. The second plot shows the distance matrix on the data set. Next, we can see the kernel matrices. The last plot of a row shows the similarity scores (encoded by color) based on $k_1$ between one fixed point (shown as a black cross) and the other points in the data set. Clearly, the kernel matrices reflect the block structures of the distance matrices, and the similarity scores between a fixed point and the other points tend to decrease as the distances to the fixed point increase. A situation like in the example of Figure 2 does not occur.

## 2.3 Landmark design

Our kernel functions are designed as to extract information from an arbitrary collection $\mathcal{S}$ of similarity triplets. However, by construction, a single triplet is useless, and what matters is the concurrent presence of two triplets: $k_1(x_a, x_b)$ is only affected by pairs of triplets answering $d(x_a, x_i) \overset{?}{<} d(x_a, x_j)$ and $d(x_b, x_i) \overset{?}{<} d(x_b, x_j)$, while $k_2(x_a, x_b)$ is only affected by pairs of triplets answering (4). Hence, when we can choose which dissimilarity comparisons of the form (1) are evaluated for creating $\mathcal{S}$ (e.g., in crowdsourcing), we should aim at maximizing the number of appropriate pairs of triplets. This can easily be achieved by means of a landmark design inspired from landmark multidimensional scaling (de Silva and Tenenbaum, 2004): We choose a small subset of landmark objects $\mathcal{L} \subseteq \mathcal{D}$. Then, for $k_1$, only comparisons of the form $d(x_i, x_j) \overset{?}{<} d(x_i, x_k)$ with $x_i \in \mathcal{D}$ and $x_j, x_k \in \mathcal{L}$ are evalu-

ated. For $k_2$, only comparisons of the form $d(x_j, x_i) \overset{?}{<} d(x_j, x_k)$ with $x_i \in \mathcal{D}$ and $x_j, x_k \in \mathcal{L}$ are evaluated. The landmark objects can be chosen either randomly or, if available, based on additional knowledge about $\mathcal{D}$ and the task at hand.

## 2.4 Computational complexity

**General $\mathcal{S}$**   A naive implementation of our kernel functions explicitly computes the feature vectors $\Phi_{k_1}(x_i)$ or $\Phi_{k_2}(x_i)$, $i = 1, \ldots, n$, and subsequently calculates the kernel matrix $K$ by means of (3) or (5). In doing so, we store the feature vectors in the feature matrix $\Phi_{k_1}(\mathcal{D}) = (\Phi_{k_1}(x_i))_{i=1}^n \in \mathbb{R}^{\binom{n}{2} \times n}$ or $\Phi_{k_2}(\mathcal{D}) = (\Phi_{k_2}(x_i))_{i=1}^n \in \mathbb{R}^{n^2 \times n}$. Proceeding this way is straightforward and simple, requiring to go through $\mathcal{S}$ only once, but comes with a computational cost of $\mathcal{O}(|\mathcal{S}| + n^4)$ operations. Note that the number of different distance comparisons of the form (1) is $\mathcal{O}(n^3)$ and hence one might expect that $|\mathcal{S}| \in \mathcal{O}(n^3)$ and $\mathcal{O}(|\mathcal{S}| + n^4) = \mathcal{O}(n^4)$. By performing (3) or (5) in terms of matrix multiplication $\Phi_{k_1}(\mathcal{D})^T \cdot \Phi_{k_1}(\mathcal{D})$ or $\Phi_{k_2}(\mathcal{D})^T \cdot \Phi_{k_2}(\mathcal{D})$ and applying Strassen's algorithm (Higham, 1990) one can reduce the number of operations to $\mathcal{O}(|\mathcal{S}| + n^{3.81})$, but still this is infeasible for many data sets. Infeasibility for large data sets, however, is even more the case for ordinal embedding algorithms, which are the current state-of-the-art method for solving machine learning problems based on similarity triplets. All existing ordinal embedding algorithms iteratively solve an optimization problem. For none of these algorithms theoretical bounds for their complexity are available in the literature, but it is widely known that their running times are prohibitively high (Heim et al., 2015; Kleindessner and von Luxburg, 2017).

**Landmark design**   If we know that $\mathcal{S}$ contains only dissimilarity comparisons involving landmark objects, we can adapt the feature matrices such that $\Phi_{k_1}(\mathcal{D}) \in \mathbb{R}^{\binom{|\mathcal{L}|}{2} \times n}$ or $\Phi_{k_2}(\mathcal{D}) \in \mathbb{R}^{|\mathcal{L}|^2 \times n}$ and reduce the number of operations to $\mathcal{O}(|\mathcal{S}| + \min\{|\mathcal{L}|^2, n\}^{\log_2(7/8)}|\mathcal{L}|^2 n^2)$, which is $\mathcal{O}(|\mathcal{S}| + |\mathcal{L}|^{1.62} n^2)$ if $|\mathcal{L}|^2 \leq n$. Note that in this case we might expect that $|\mathcal{S}| \in \mathcal{O}(|\mathcal{L}|^2 n)$.

In both cases, whenever the number of given similarity triplets $|\mathcal{S}|$ is small compared to the number of all different distance comparisons under consideration, the feature matrix $\Phi_{k_1}(\mathcal{D})$ or $\Phi_{k_2}(\mathcal{D})$ is sparse with only $\mathcal{O}(|\mathcal{S}|)$ non-zero entries and methods for sparse matrix multiplication decrease computational complexity (Gustavson, 1978; Kaplan et al., 2006).

## 3 Related work

Similarity triplets are a special case of answers to the general dissimilarity comparisons $d(A, B) \overset{?}{<} d(C, D)$, $A, B, C, D \in \mathcal{X}$. We refer to any collection of answers to these general comparisons as ordinal data. In recent years, ordinal data has become popular in machine learning. Among the work on ordinal data in general (see Kleindessner and von Luxburg, 2014, 2017, for references), similarity triplets have been paid particular attention: Jamieson and Nowak (2011) deal with the question of how many similarity triplets are required for uniquely determining an ordinal embedding of Euclidean data. This work has been carried on and generalized by Jain et al. (2016). Algorithms for constructing an ordinal embedding based on similarity triplets (but not on general ordinal data) are proposed in Tamuz et al. (2011), van der Maaten and Weinberger (2012), Amid et al. (2016), and Jain et al. (2016). Heikinheimo and Ukkonen (2013) present a method for medoid estimation based on statements "Object $A$ is the outlier within the triple of objects $(A, B, C)$", which correspond to the two similarity triplets $d(B, C) < d(B, A)$ and $d(C, B) < d(C, A)$. Ukkonen et al. (2015) use the same kind of statements for density estimation and Ukkonen (2017) uses them for clustering. Wilber et al. (2014) examine how to minimize time and costs when collecting similarity triplets via crowdsourcing. Producing a number of ordinal embeddings at the same time, each corresponding to a different dissimilarity function based on which a comparison (1) might have been evaluated, is studied in Amid and Ukkonen (2015). In Heim et al. (2015), one of the algorithms by van der Maaten and Weinberger (2012) is adapted from the batch setting to an online setting, in which similarity triplets are observed in a sequential way, using stochastic gradient descent. In Kleindessner and von Luxburg (2017), we propose algorithms for medoid estimation, outlier detection, classification, and clustering based on statements "Object $A$ is the most central object within $(A, B, C)$", which comprise the two similarity triplets $d(B, A) < d(B, C)$ and $d(C, A) < d(C, B)$. Finally, Haghiri et al. (2017) study the problem of efficient nearest neighbor search based on similarity triplets. There

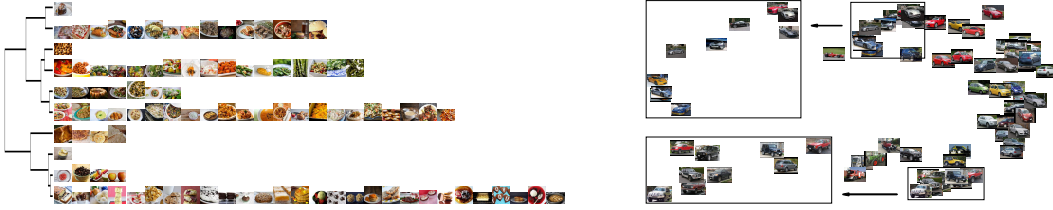

Figure 4: Best viewed magnified on screen. Left: Clustering of the food data set. Part of the dendrogram obtained from complete-linkage clustering using $k_1$. Right: Kernel PCA on the car data set based on the kernel function $k_2$.

is also a number of papers that consider similarity triplets as side information to vector data (e.g., Schultz and Joachims, 2003; McFee and Lanckriet, 2011; Wilber et al., 2015).

# 4 Experiments

We performed experiments that demonstrate the usefulness of our kernel functions. We first apply them to three small image data sets for which similarity triplets have been gathered via crowdsourcing. We then study them more systematically and compare them to an ordinal embedding approach in clustering tasks on subsets of USPS and MNIST digits using synthetically generated triplets.

## 4.1 Crowdsourced similarity triplets

In this section we present experiments on real crowdsourcing data that show that our kernel functions can capture the structure of a data set. Note that for the following data sets there is no ground truth available and hence there is no way other than visual inspection for evaluating our results.

**Food data set**   We applied the kernelized version of complete-linkage clustering based on our kernel function $k_1$ to the food data set introduced in Wilber et al. (2014). This data set consists of 100 images[2] of a wide range of foods and comes with 190376 (unique) similarity triplets, which contain 9349 pairs of contradicting triplets. Figure 4 (left) shows a part of the dendrogram that we obtained. Each of the ten clusters depicted there contains pretty homogeneous images. For example, the fourth row only shows vegetables and salads whereas the ninth row only shows fruits and the last row only shows desserts. To give an impression of accelerated running time of our approach compared to an ordinal embedding approach: computation of $k_1$ or $k_2$ on this data set took about 0.1 seconds while computing an ordinal embedding using the GNMDS algorithm (Agarwal et al., 2007) took 18 seconds (embedding dimension equaling two; all computations performed in Matlab—see Section 4.2 for details; the embedding is shown in Figure 9 in Section A.1 in the supplementary material).

**Car data set**   We applied kernel PCA (Schölkopf et al., 1999) based on our kernel function $k_2$ to the car data set, which we have introduced in Kleindessner and von Luxburg (2017). It consists of 60 images of cars. For this data set we have collected statements of the kind "Object $A$ is the most central object within $(A, B, C)$", meaning that $d(B, A) < d(B, C)$ and $d(C, A) < d(C, B)$, via crowdsourcing. We ended up with 13514 similarity triplets, of which 12502 were unique. The projection of the car data set onto the first two kernel principal components can be seen in Figure 4 (right). The result looks reasonable, with the cars arranged in groups of sports cars (top left), ordinary cars (middle right) and off-road/sport utility vehicles (bottom left). Also within these groups there is some reasonable structure. For example, the race-like sports cars are located near to each other and close to the Formula One car, and the sport utility vehicles from German manufacturers are placed next to each other.

**Nature data set**   We performed similar experiments on the nature data set introduced in Heikinheimo and Ukkonen (2013). The results are presented in Section A.2 in the supplementary material.

We would like to discuss a question raised by one of the reviewers: in our setup (see Section 1), we assume that similarity triplets are noisy evaluations of dissimilarity comparisons (1), where $d$ is some fixed dissimilarity function. This leads to our (natural) way of dealing with contradicting similarity triplets as described in Section 2. In a different setup one could drop the dissimilarity function $d$ and consider similarity triplets as elements of some binary relation on $\mathcal{D} \times \mathcal{D}$ that is not necessarily transitive or antisymmetric. In the latter setup it is not clear whether our way of dealing with contradicting triplets is the right thing to do. However, we believe that the experiments of this section show that our setup is valid in a wide range of scenarios and our approach works in practice.

## 4.2 Synthetically generated triplets

We studied our kernel functions with respect to the number of input similarity triplets that they require in order to produce a valuable solution in clustering tasks. We found that in the scenario of a general collection $\mathcal{S}$ of triplets our approach is highly superior compared to an ordinal embedding approach in terms of running time, but on most data sets it is inferior regarding the required number of triplets. The full benefit of our kernel functions emerges in a landmark design. There our approach can compete with an embedding approach in terms of the required number of triplets and is so much faster as to being easily applicable to large data sets to which ordinal embedding algorithms are not. In this section we want to demonstrate this claim. We studied $k_1$ and $k_2$ in a landmark design by applying kernel $k$-means clustering (Dhillon et al., 2001) to subsets of USPS and MNIST digits, respectively. Collections $\mathcal{S}$ of similarity triplets were generated as follows: We chose a certain number of landmark objects uniformly at random from all objects of the data set under consideration. Choosing $d$ as the Euclidean metric, we created answers to all possible distance comparisons with the landmark objects as explained in Section 2.3. Answers were incorrect with some probability $0 \leq ep \leq 1$ independently of each other. From the set of all answers we chose triplets in $\mathcal{S}$ uniformly at random without replacement. We compared our approach to an ordinal embedding approach with ordinary $k$-means clustering. We tried the GNMDS (Agarwal et al., 2007), the CKL (Tamuz et al., 2011), and the t-STE (van der Maaten and Weinberger, 2012) embedding algorithms in the Matlab implementation made available by van der Maaten and Weinberger (2012). In doing so, we set all parameters except the embedding dimension to the provided default parameters. The parameter $\mu$ of the CKL algorithm was set to 0.1 since we observed good results with this value. Note that in these unsupervised clustering tasks there is no immediate way of performing cross-validation for choosing parameters. We compared to the embedding algorithms in two scenarios: in one case they were provided the same triplets as input as our kernel functions, in the other case (denoted by the additional "rand" in the plots) they were provided a same number of triplets chosen uniformly at random with replacement from all possible triplets (no landmark design) and incorrect with the same probability $ep$. For further comparison, we considered ordinary $k$-means applied to the original point set and a random clustering. We always provided the correct number of clusters as input, and set the number of replicates in $k$-means and kernel $k$-means to five and the maximum number of iterations to 100. For assessing the quality of a clustering we computed its purity (e.g., Manning et al., 2008), which measures the accordance with the known ground truth partitioning according to the digits' values. A high purity value indicates a good clustering. Note that the limitation for the scale of our experiments only comes from the running time of the embedding algorithms and not from our kernel functions. Still, in terms of the number of data points our experiments are comparable or actually even superior to all the papers on ordinal embedding cited in Section 3. In terms of the number of similarity triplets per data point, we used comparable numbers of triplets.

**USPS digits**    We chose 1000 points uniformly at random from the subset of USPS digits 1, 2, and 3. Using 15 landmark objects, we studied the performance of our approach and the ordinal embedding approach as a function of the number of input triplets. The first and the second row of Figure 5 show the results (average over 10 runs of an experiment) for $k_1$. The results for $k_2$ are shown in Figure 7 in Section A.1 in the supplementary material. The first two plots of a row show the purity values of the various clusterings for $ep = 0$ and $ep = 0.3$, respectively. The third and the fourth plot show the corresponding time (in sec) that it took to compute our kernel function or an ordinal embedding. We set the embedding dimension to 2 (1st row) or 10 (2nd row). Based on the achieved purity values no method can be considered superior. Our kernel function $k_2$ performs slightly worse than $k_1$ and the ordinal embedding algorithms. The GNMDS algorithm apparently cannot deal with the landmark triplets at all and yields the same purity values as a random clustering when provided with the landmark triplets. Our approach is highly superior regarding running time. The running times of the

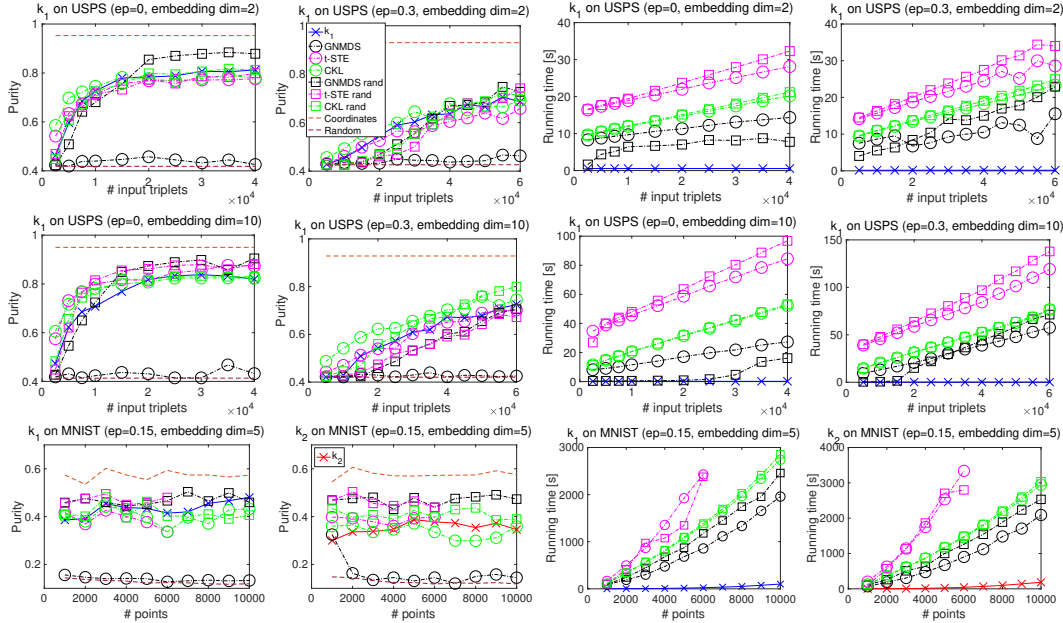

Figure 5: **1st & 2nd row (USPS digits for $k_1$):** Clustering 1000 points from USPS digits 1, 2, and 3. Purity and running time as a function of the number of input triplets. **3rd row (MNIST digits):** Clustering subsets of MNIST digits. Purity and running time as a function of the number of points.

ordinal embedding algorithms depend on the embedding dimension and $ep$ and in these experiments the dependence is monotonic. All computations were performed in Matlab R2016a on a MacBook Pro with 2.9 GHz Intel Core i7 and 8 GB 1600 MHz DDR3. In order to make a fair comparison we did not use MEX files or sparse matrix operations in the implementation of our kernel functions.

**MNIST digits** We studied the performance of the various methods as a function of the size $n$ of the data set with the number of input triplets growing linearly with $n$. For $i = 1, \ldots, 10$, we chose $n = i \cdot 10^3$ points uniformly at random from MNIST digits. We used 30 landmark objects and provided $150n$ input similarity triplets. The third row of Figure 5 shows the purity values of the various methods for $k_1$ / $k_2$ (1st / 2nd plot) and the corresponding running times (3rd / 4th plot) when $ep = 0.15$. The embedding dimension was set to 5. A spot check suggested that setting it to 2 would have given worse results, while setting it to 10 would have given similar results, but would have led to a higher running time. We computed the t-STE embedding only for $n \leq 6000$ due to its high running time. It seems that GNMDS with random input triplets performs best, but for large values of $n$ our kernel function $k_1$ can compete with it. For 10000 points, computing $k_1$ or $k_2$ took 100 or 180 seconds, while even the fastest embedding algorithm ran for 2000 seconds. For further comparison, Figure 8 in Section A.1 in the supplementary material shows a kernel PCA embedding based on $k_1$ ($150n$ landmark triplets) and a 2-dim GNMDS embedding ($150n$ random triplets) of $n = 20000$ digits. Here, computation of $k_1$ took 900 seconds, while GNMDS ran for more than 6000 seconds.

## 5  Conclusion

We proposed two data-dependent kernel functions that can be evaluated when given only an arbitrary collection of similarity triplets for a data set $\mathcal{D}$. Our kernel functions can be used to apply any kernel method to $\mathcal{D}$. Hence they provide a generic alternative to the standard ordinal embedding approach based on numerical optimization for machine learning with similarity triplets. In a number of experiments we demonstrated the meaningfulness of our kernel functions. A big advantage of our kernel functions compared to the ordinal embedding approach is that our kernel functions run significantly faster. A drawback is that, in general, they seem to require a higher number of similarity triplets for capturing the structure of a data set. However, in a landmark design our kernel functions can compete with the ordinal embedding approach in terms of the required number of triplets.

**Acknowledgements**

This work has been supported by the Institutional Strategy of the University of Tübingen (DFG, ZUK 63).

## Footnotes

[2]According to Wilber et al., the data set contains copyrighted material under the educational fair use exemption to the U.S. copyright law.

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
