[Reviews · NeurIPS 2017]

Reviewer 1



The authors propose two kernel functions based on comparing triplets of data examples. The first kernel function is based on ranking of all the other objects with respect to the objects being compared. The second function is based on comparing whether the compared objects rank similarly with respect to the other objects. The idea is interesting and it is potentially impactful to create such “ordinal” kernels, where the data are not embedded in metric spaces. Comments: 1. The authors should provide their main claims somewhere and demonstrate it well in the paper. Why should anybody use a triplet comparison kernel? From what I understand, with triplet comparison kernel, less number of triplets are sufficient to capture the same information that a full pairwise similarity matrix can capture (sec. 2.2). It is also faster. Is that correct? 2. How is the first kernel (k_1) and the one proposed in (Jiao & Vert 2015) different? 3. In Figure 3, can you quantify how well the proposed kernel performs with respect to computing all pairwise similarities? The synthetic datasets are not hard, and it may also be useful to demonstrate some harder case? 4. The kernels seem to have time advantages compared to other embedding approaches such as GNMDS (from the experiments in Sec. 4.2). However in terms of clustering performance they are outperformed by embedding approaches (the authors claim that the performance is close for all algorithms). How many landmark points were chosen for computing the kernels? Can you comment on the size of landmark set vs. running times? 5. Also, in terms of running times, can you comment whether the reason for the slow performance of the embedding algorithms is the inherent computational complexity? Could it be an implementation issue? 6. The authors should attempt to provide results that are legible and easy to read. If there is lack of space, supplementary material can be provided with the rest of results. Update ------- The authors have answered several of my comments satisfactorily.

Reviewer 2



Paper Summary: The authors propose a methodology to define a similarity kernel function from ordinal data (triplet based comparisons). They estimate the (dis)similarity score between two objects x1 and x2 by comparing rankings induced the objects that are estimated from the ordinal data. Experiments on benchmark datasets show that their proposed technique yields comparable quality results but with significant speedups. Review: Overall I like the idea of estimating kernel functions from ordinal data. To best of my knowledge this has not been done before and I believe that the intuitive idea proposed in this paper can help practitioners do standard kernel-based learning tasks from ordinal data. I hope that the authors do the following: - include all the ‘omitted’ figures and experiments on a more diverse set of benchmarks in the appendix / supplementary materials. - provide the demo code - more visualizations like Figure 4 These would help make the message of the paper even stronger.

Reviewer 3



The authors propose kernels on the top of non-metric, relatively ordered data sources. In the construction they define explicit feature spaces to represent the kernels, thus those kernels could be relatively easily applied for example to describe outputs in a structured learning environment as well. The geometry of the feature space is also discussed to illuminate the potential behavior which can help in the interpretation of the relationships expressible by these type of kernels. The paper is clearly, and concisely written. The introduction of the underlying problem can be well understood. The definition of the kernels via explicit feature space is significant advantage against other kernels built up on a graph structure. Firstly it seems the data model is very restricted, but this type of approach to learning from non-metric data is inherently allows to tackle with highly nonlinear problems, for example learning on a manifold where only local relations are available. The handling of contradictions might be oversimplified since they could be also produced by non-acyclic graphs and not only by statistical noise. Perhaps a better approach could be that where the category of the underlying order structures are defined more explicitly. In that case methods dealing with the different type of contradictions can be developed.